# Analysis of Supporting Factors Associated with Exclusive Breastfeeding Practice in the Urban Setting during the COVID-19 Pandemic

**DOI:** 10.3390/children9071074

**Published:** 2022-07-19

**Authors:** Agrina Agrina, Dedi Afandi, Suyanto Suyanto, Erika Erika, Yulia Irvani Dewi, Siska Helina, Dita Pramita, Nanda Safira

**Affiliations:** 1Faculty of Nursing, Universitas Riau, Pekanbaru 28293, Indonesia; agrina@lecturer.unri.ac.id (A.A.); erika@lecturer.unri.ac.id (E.E.); yulia.irvanidewi@lecturer.unri.ac.id (Y.I.D.); 2Faculty of Medicine, Universitas Riau, Pekanbaru 28293, Indonesia; dediafandi4n6@gmail.com; 3Politeknik Kesehatan Riau, Pekanbaru 28156, Indonesia; siska@pkr.ac.id; 4PMC Health School, Pekanbaru 28126, Indonesia; ditarhmn@gmail.com; 5Epidemiology Department, Prince of Songkla University, Hatyai 90110, Thailand; safiraa25@gmail.com

**Keywords:** breastfeeding support, COVID 19 pandemic, breastfeeding practice

## Abstract

Breastfeeding mothers have had limited access to breastfeeding support throughout the COVID-19 pandemic. This study aims to investigate breastfeeding practices during the COVID-19 period and to determine the factors associated with supporting exclusive breastfeeding. A sequential explanatory mixed methods approach was adopted, including a quantitative method in the first phase and qualitative method in the second phase. Mothers whose babies were aged over 6 months to 24 months old from July to September 2021 in Pekanbaru City were selected as research subjects. Data analysis was performed with multivariate and deductive content analysis. Of 156 participants, 97 mothers (62.2%) exclusively breastfed their babies. Of those, mothers who delivered exclusive breastfeeding worked less than eight hours per day, were aged 17–25 and had low education. Though by using exclusive breastfeeding practice as a reference, associated supports, including emotional, instrumental, appraisal and information regarding exclusive breastfeeding practice were insignificant; however, mothers who practice exclusive breastfeeding had higher information support. During the COVID-19 pandemic period, the informational support factor was found to be important to achieve the successful exclusive breastfeeding practice.

## 1. Introduction

Supporting mothers to continue breastfeeding during the COVID-19 pandemic is a public health concern. It is globally recommended that breastfeeding should be continued during the pandemic to increase the health and immunity of babies [1]. Breast milk is the finest source of nutrients, which includes a variety of nutrients necessary for an infant’s growth and development, as well as substances that help to boost the immune system [2].

Exclusive breastfeeding for the first six months of a new-born’s life reduces the risk of infectious diseases in infants, protects them from chronic diseases in adulthood and increases intelligence scores [3,4,5,6]. Therefore, breastfed new-borns were found to result in a reduction in infant death and morbidity [7]. The World Health Organization (WHO) recommends that every infant breastfeed exclusively for the first six months of life, followed by supplemental feeding until the age of two years to see the best benefits. 

The WHO has planned to increase the rate of exclusive breastfeeding in the first six months to at least 50% by the year of 2025; however, the percentage of infants under six months who are exclusively breastfed is only 40% globally [8]. Indonesia has implemented the regulations to support complete breastfeeding practice; however, it ranked 49th out of 51 countries in 2012 based on World Breastfeeding data for breastfeeding conditions, with a breastfeeding rate of 27.5%, and only 42% of children under six months are exclusively breastfed aaccording to the 2012 Indonesian Demographic and Health Survey respectively [9]. A recent study found that half of Indonesian mothers who had babies aged 6–12 months stopped exclusively breastfeeding during the first month of the baby’s life and the breastfeeding lasted an average of 2.03 months [10]. This condition could increase the morbidity and mortality rates due to infants failing to obtain the best food in their early life.

The progression of COVID-19 cases increased during the early stage of the Covid pandemic, and it appears that COVID-19 will remain endemic [11]. On the other hand, during the COVID-19 pandemic, the attention of health services was more geared toward resolving the COVID-19 problem, which might have led to mothers not receiving effective breastfeeding support. Furthermore, the COVID-19 phenomenon reshaped to new adaptations, such as working from home, lockdown restrictions and social distancing rules, which has created a variety of obstacles for breastfeeding women, including the support available to them as well as adequate support from the community level [12].

The imposition of lockdown and home confinement led to a decrease in exclusive breastfeeding during the COVID-19 pandemic especially among working mothers in urban areas [13]. Factors, such as the maternal age, socioeconomic status, parity, psychosocial factors, caesarean delivery, perceptions of inadequate breastfeeding, occupation, counselling on feeding during postnatal care and the length of time the mother gave birth were predictors of the duration of breastfeeding [14,15,16].

Studies have also shown that the support provided to breastfeeding mothers is an essential factor in successful breastfeeding in Indonesia [17]. However, studies on breastfeeding practices, the support provided during the COVID-19 pandemic, including the type of breastfeeding support (emotional, informational, instrumental and appraisal support), and the dominant support that affects the success of exclusive breastfeeding is still limited. Such information is potentially useful in the planning and decision-making processes of health programs and policy makers. Hence, this study aimed to assess the breastfeeding practices and to determine the associated factors with supporting exclusive breastfeeding during the COVID-19 pandemic.

## 2. Materials and Methods

This study obtained using a mixed-method with a sequential explanatory and cross sectional design approaches. Purposive sampling was used to collect quantitative data from 156 breastfeeding mothers with babies older than six months old from two public health facilities between August and September 2021. We used the Slovin formula with 95% confidence interval for sample size study estimation. 

The Family Support Questionnaire (FSQ) originally was developed by Biswass [18] to measure family support factors. Moreover, it was translated and modified by Vita sari [19], which consists of 20 items included emotional, instrumental, informational and assessment supports. The FSQ questionnaire employs a Likert scale measurement, with 1 indicating never, 2 indicating sometimes, 3 indicating often, 4 indicating very often and 5 indicating always, for a total score of 20–100.

The qualitative study using the semi-structured in-depth interviews enlisted the participation of 12 informants who were breastfeeding mothers with eligible new-borns. Exclusive breastfeeding in this study means that the baby provided only mothers breast milk from birth to 6 months without any additional drinks and food. Research data were collected by fourth year students of the Faculty of Nursing, University of Riau who have been trained as research data collectors. 

First, the quantitative data was analysed using chi-square test for independence to test the association between sociodemographic variables and breastfeeding practice patterns, and then the multinomial logistic regression analysis was performed to find whether the independent factors included the breastfeeding support were factors associated towards exclusive breastfeeding practice. The outcome in this study has therefore three nominal categories, which were a combination of breastfeeding and formula milk (mixed), breastfeeding and infant formula. Independent variables with the p-value less than 0.05 were assigned as significantly associated with the outcome variable.

The qualitative data was analysed using thematic data analysis. Both quantitative and qualitative results were finally integrated. The ethical test was issued by the ethics committee with the number 283/U.19.5.1.8/KEPK.FKp/2021.

## 3. Results

### 3.1. Quantitative Result

At the study locations, there were 156 breastfeeding mothers as samples. The study had babies with ages ranging from 8 to 15 months with most of the mothers aged above 25 years old (59.6%). More than a half of the mothers had given only breast milk before the babies were six months old (62.2%). The sociodemographic background had an association with breastfeeding practice among the study participants. These sociodemographic variables were a low education level (junior to middle high school) and working 8 h or less in a day (including housewives) as shown in the Table 1. Mothers who had a high education level were more likely to mix breast milk with formula milk given to their babies (70.6%), while nearly all of mothers who worked less than 8 h gave only breast milk to their babies (90.7%).

Table 2 explains the association between characteristic factors and exclusive breastfeeding practice using multinomial logistic regression analysis. Considering exclusive breastfeeding practice as a reference, the demographic variables, such as mothers aged more than 33 years old, babies age 15 to 24 months, high education level and working hour more than 8 h, were more likely to provide a combination of breastfeeding and formula milk (mixed group). Mothers with children aged 15 to 24 months were likely to provide infant formula group.

There was inadequate evidence that supporting factors, such as emotional, informational, instrumental and appraisal, were associated with the decision to provide exclusive breastfeeding among mixed and infant formula groups. However, mothers in the high informational group were more likely to nurse their baby exclusively with breastmilk compared to those who mixed both breastfeeding and formula milk, respectively (RRR = 0.43; 95%CI = 0.96–2.82).

### 3.2. Qualitative Result

According to in-depth interviews, participants had a number of challenges when breastfeeding their babies, such as being late on early initiation of breastfeeding (IMD) and the problem of breast milk that did not come out. Here are the participants’ expressions:
*“Saya udah mencoba memberikan ASI, bayi diletakkan di dada… itupun cuman sebentar sekitar 5 menit lah, bayinya cuman menjilat-jilat aja, memang gak keluar (ASI)…”*(P2)
**“I’ve tried to feed breast milk, the baby is placed on the chest… it only lasted for about 5 min, the baby just licked it, the breast milk just didn’t come out…”**

Breastfeeding issues were not only during the IMD procedure, mothers were also experiencing breastfeeding challenges dependent on the infant and mother, which included not enough breast milk or low milk supply, soreness experience, pain and swelling. Here are the responses of the participants from mother’s perspectives:
*“… Udah dipencet-pencet payudaranya, ASInya gak ada pas pertama lahiran”*(P2)
**“…The breasts have been squeezed but the breast milk is not there at the beginning of giving birth”**
*“gak ada masalah dengan payudara, putingnya ada, bersih, cuman bengkak, apakah karena banyinya sudah kenyang diberi susu (susu formula) sama bidannya, kayak malas gitu menyusu”*(P4)
**“There is no issue with the breasts, just the nipples are swollen, it might due to the baby is too tired to breastfeed maybe because the baby has already given the milk (formula milk) by the midwife.”**
*“Mimiknya (payudara)… nyeri terus kalau dia nyusu tu, ada luka, dianya (bayi) kuat mimik.. ngisap ujungnya aja, …akhirnya awak jadi takut kalau dia nyusu”*(P9)
***“My breasts… when the baby is breastfeeding, there is a lesion, baby sucks aggressively. only sucks the tip, …in the end when he suckles, I feel anxious.”***
*“… dia gak tidur-tidur, nangis-nangis terus, ha stress lah awak, anak awak ndak tidur-tidur, capek juga, awak ndak tidur kan… itulah air, air susu tu kurang…”*(P12)
**“… The infant does not sleep and constantly cries. I’m really stressed. My child isn’t sleeping and I’m exhausted, therefore I don’t sleep. Finally, the milk gets scarce…”**

Breastfeeding problems arise not only from the mother’s perspective but also from the baby’s standpoint, such as a baby who has difficulty sucking. These following expressions from participants were presented below:
*“pas bayinya menghisap itu kan kenak, kenak gusinya gitu pentilnya itu…”*(P1)
**“when the baby suckes, the baby’s gums force against the nipple…”**
*“pas N cari-cari, gara-gara gak tepat masuk ke mulut katanya, apa yang bulatan tu gak masuk ke bibir”*(P3)
**“when breastfeeding, the baby’s mouth does not suck properly on the nipple”**

Participants received various forms of support while breastfeeding. All participants briefly stated that their husbands and families provided them with great support. Husbands provide significant support, particularly in early breastfeeding initiation, which includes paying attention to the wife’s food consumption, massaging the wife’s back and offering help. During breastfeeding, families may offer support and assistance. The participants received family assistance, which included guidance to pay attention to the food consumption, assist the mother’s work and dietary items that can aid in the production of a large amount of breast milk. Here are the reactions of the participants with their husband and family supports:
*“iya, jadi kami juga ada dipijat di punggung gitu lihat di* YouTube… *suami awak peragakan langsung gimana gerakannya, memang terasa enak, katanya memperbanyak air susu gitu”*(P2)
**“Yes, I had my back massaged. My husband discovered the technique on YouTube… My spouse performed the steps. I feel better, and it has the potential to enhance milk supply.”**
*“iya mendukung bapaknya, dia belikan makanan yang memperbanyak air susu… makan tapai (tape singkong) kata orang kan banyak air susu, tempe, tahu, sayur… sayur katuk, bayam hampir tiap hari”*(P10)
**“My husband supports me to breastfeed the baby. He buys food to booster breast milk… People believe that eating tapai (fermented cassava) helps enhance breast milk production, consume tempe, tofu, vegetables… such as katuk, spinach almost every day”**
*“… memang iya kalau sama neneknya waktu tu kan neneknya yang ngasih nyediain makan kan selama empat puluh harikan, memang dipantang kan gak dikasih cabe”*(P1)
**“… it’s true that grandmother served meals for forty days (for mother) during which chilli consumption was prohibited”**
*“mama sangat perhatian, ngasih saran makan aja yang banyak untuk bayinya, sayuran, buah-buahan… cuman kan kata mama jangan makan cabe dulu”. “disuruh makan yang pahit-pahit, jangan yang manis-manis… iya biar cepat kuat aja”*(P4)
**“My mother is much caring, offering tips on how to feed my baby a lot of veggies and fruits… Mom urged me not to eat chilies first and to eat the bitter ones instead of the sweet ones so I could grow strong soon”**
*“… gak boleh pegang sapu, gak boleh kerja berat-berat, perutnya harus diikat biar jangan apa… kayak apa ya, biar gak turun peranakannya (rahim), ASI nya biar banyak”*(P7)
**“… not allowed to sweep the floor, not allowed to do heavy work, I have to tighten my stomach to keep uterine prolapse from coming down, then I can have breast milk excessively”**
*“… mamak bikin kan ikan baung eh ikan apa ikan apa namanya ikan gabus direbus…cepat keringnya (luka jalan lahir)… “iya makan daun katu (katuk), sayur bayam, katanya banyak ASI, disayur bening aja”*(P12)
**“… my mother cooks boiled cork fish to help the birth canal wound heal rapidly.” “My mother also suggests eating katu (katuk) leaves and spinach to ensure a plentiful supply of breast milk”**

On the other hand, the healthcare workers were more likely to not provide any support. Some participants also mentioned that the healthcare workers did not assist with further information about breastfeeding after giving birth. Here are the participants’ expressions:
*“… gak ada informasi apa-apa bu, dia hanya nolong lahiran, sehabis tu membersihin bayinya…”*(P4)
**“I didn’t get any information from the midwife. she just helped the delivery, after that she cleaned the baby…”**
*“kalau menjelaskan tentang manfaat ASI ndak ada, cuman disuruh nyusuin anaknya aja… pas sakit pada puting, hanya dikasih salap… ndak pa pa itu katanya…”*(P5)
**“There was no explanation regarding the benefits of breastfeeding; I was just instructed to breastfeed the baby. When my nipples hurt, she just advised ointment…”**

## 4. Discussion

This mixed-method study was conducted to assess the relationship between breastfeeding practices and the factors that support exclusive breastfeeding during the COVID-19 pandemic in Indonesia. More than half of mothers with babies aged six months and older delivered breast milk for the first six months of their child’s life.

Mothers with children aged 15 to 24 months were shown to be more likely to provide their babies with a combination of breastfeeding and formula milk compared to infant formula and breastfeeding milk. The older age of mothers aged more than 33 years old, a high education level and working more than 8 h per day showed the same evidence with mothers having children aged 15 to 24 months. We found insufficient evidence on any supporting factors regarding exclusive breastfeeding compared to infant formula milk or the combination of both to their babies.

The COVID-19 pandemic has caused significant changes in lifestyle and disruption in various fields, including in the health care sector [20]. As shown in our study result, over half of mothers breastfed their babies and, during the COVID-19 pandemic in 2020, the number of exclusive breastfeeding cases increased [21]. In addition, that the limitation of operational health services related to children under-five does not discourage lactation behaviour in Indonesia. However, behind this success, several groups experience failure in deliver exclusive breastfeeding.

Various factors may influence the mother’s inability to give exclusive breastfeeding, such as education, work and other support [22]. Other studies has also found that a low education level of the mother affects the success of exclusive breastfeeding due to many misconceptions [23,24]. Those with a higher education level are better at making decisions, one of which is exclusive breastfeeding for babies [25]. However, in this study, respondents with higher education were more likely to give mixed milk (breast milk and formula milk).

This is because respondents with higher education are more frequently employed. Occupation is a predictor of failure in exclusive breastfeeding. Moreover, the higher economic status tends not to give exclusive breastfeeding. This was supported by the affordability to buy extra food, such as milk formula that could be given to infants in addition to breast milk. The study also revealed that the welfare of the family played an important role in the breastfeeding habit [26]. Infants from families with higher economic status rarely had exclusive breastfeeding because of exposure to formula milk and financial support [27].

Apart from the education factor, the working hours were identified as affecting exclusive breastfeeding practice. Working mothers could decrease the opportunity for exclusive breastfeeding in which the mother worked all the time were 1.54-times more likely not to give exclusive breastfeeding than mothers who did not work [28]. Likewise, a working mother is likely to breastfeed with a short duration compared to a woman who is unemployed or part time worker [29]. Study in Padang, Indonesia found that on average, the nursing mother who works in the office every day leaves her baby at home at least 10 hour every day, which is calculated from the time mother goes out from home to the workplace until returning back home [30].

Working long hours weakened mother-to-child intuition, and physical tiredness might disrupt the let-down reflex and reduce milk production in women [31]. This condition may affect the duration and amount of exclusive breastfeeding given to children. Since the short work breaks, the mother lacks desire to give expressed breast milk. As a result, numerous mothers administer complementary food (MP-ASI) to their babies when they are less than six months old.

The high percentage of exclusive breastfeeding in this research was attributed to the fact that the majority of mothers did not work or worked for less than 8 hour in the COVID-19 period, whereas the last study also showed that mothers who worked more than 7 hour per a day in urban areas during COVID-19 pandemic were likely to stop breastfeeding earlier [13]. Furthermore, breastfeeding mothers who worked from home (WFH) during the Limited Outdoor Activity Policy or Regional Lockdown (PSBB) had an excellent opportunity to continue breastfeeding their babies.

The success of exclusive breastfeeding during the pandemic cannot be separated from the husband’s support for breastfeeding mothers; this is because the husband’s assessment of support is the most dominant component of exclusive breastfeeding. There are two positive categories of husband’s support: instrumental and assessment support and two other husbands’ supports, namely emotional and informational support [32]. The support of the closest person is an essential component in the success of exclusive breastfeeding. 

Mothers will be more capable and confident when given support by those closest to them when breastfeeding their babies [33] and particularly with the support of their partners’ verbal encouragement and husbands who participated in breastfeeding activities. Mothers who were actively supported by their husbands had a higher Breastfeeding Self-Efficacy Scale (BSES) than mothers who were not supported by their husbands [34]. In Zambia, a similar study was also conducted, which showed that support for breastfeeding mothers was significantly associated with exclusive breastfeeding [35].

During the COVID-19 pandemic, breastfeeding mothers faced various challenges and changes in their lives, including breastfeeding practices. Such policies have led to substantial changes for breastfeeding women, including decreased in-person support from extended family and professional services, increased virtual assistance, employment changes and additional duties at home as a result of company and school closures. Participants who believed they received aid from a number of sources, including their spouses and extended relatives, demonstrated this. 

Although assistance from health personnel was not received, support from family members, particularly informational support, was extremely significant in this study. It seems that online health service consultations during the COVID-19 pandemic were beneficial and effective [36]. Given that there were differences in the skills of breastfeeding mothers who received it, such informational support obtained online has been essential during the COVID-19 pandemic in helping mothers to tackle breastfeeding-related problems [37]. Another study also shown that breastfeeding informational support through health education can increase breastfeeding skill [38].

This study provides valuable information on breastfeeding practices in urban areas in Indonesia during the COVID-19 pandemic. However, the study has several limitations. It is difficult to meet breastfeeding mothers because most mothers are afraid to meet in person face-to-face for data collection by questionnaires or interviews. Future researchers need to evaluate the effectiveness of using various information media needed by breastfeeding mothers both during the pandemic and after the pandemic so that breastfeeding information needs can be met.

## 5. Conclusions

Factors that may influence exclusive breastfeeding were the mother’s age being more than 33 years old, having children aged 15 to 24 months, having a high education level and working more than eight hours. All supporting factors had no evidence toward exclusive breastfeeding practices among mothers. Hence, the informational support factor was found to be beneficial for successfully exclusive breastfeeding during the COVID-19 pandemic period.

## Figures and Tables

**Table 1 children-09-01074-t001:** Participant characteristics stratified by breastfeeding practice patterns.

Characteristics	Formula Milk (%)	Breastfeeding (%)	Mixed ^a^ (%)	*p* Value
**Child age group**				<0.01 *
6–8 month	1 (4)	29 (29.9)	2 (5.9)	
8–15 month	10 (40)	62 (63.9)	12 (35.3)	
15–24 month	14 (56)	6 (6.2)	20 (58.8)	
**Mother age group**				0.08
17–25 y.o	1 (4)	19 (19.6)	6 (17.6)	
25–33 y.o	16 (64)	57 (58.8)	14 (41.2)	
33–47 y.o	8 (32)	21 (21.6)	14 (41.2)	
**Education level**				<0.01 *
Low	9 (36)	66 (68)	10 (29.4)	
High	16 (64)	31 (32)	24 (70.6)	
**Working hour**				<0.01 *
≥8 h per day	6 (24)	9 (9.3)	16 (47.1)	
<8 h per day ^b^	19 (76)	88 (90.7)	18 (52.9)	
**Emotional support**				0.18
Low	4 (16)	43 (44.3)	4 (11.8)	
High	21 (84)	54 (55.7)	30 (88.2)	
**Informational Support**				<0.01 *
Low	5 (20)	29 (29.9)	5 (14.7)	
High	20 (80)	68 (70.1)	29 (85.3)	
**Instrumental Support**				<0.02 *
Low	3 (12)	29 (29.9)	5 (14.7)	
High	22 (88)	68 (70.1)	29 (85.3)	
**Appraisal support**				<0.01 *
Low	2 (8)	32 (33)	3 (8.8)	
High	23 (92)	65 (67)	31 (91.2)	

^a^ Both breast and formula milk. ^b^. Including housewives. *. Significantly statistic *p* value less than 0.05.

**Table 2 children-09-01074-t002:** Multivariable adjusted multinomial logistic regression analysis for the association of independent factors towards exclusive breastfeeding practice.

Variable	Mixed vs. Breastfeeding		Infant Formula vs. Breastfeeding	
	RRR	CI	RRR	CI
**Child Age**				
6–8 month	0.37	0.06–2.25	0.21	0.02–1.88
8–15 month	1		1	
15–24 month	20.2	5.45–74.7 *	13.0	3.62–46.4 *
**Mother age**				
17–25 y.o	1.89	0.37–9.73	0.26	0.03–2.65
25–33 y.o	1		1	
33–47 y.o	3.76	1.10–12.9 *	1.92	0.57–6.53
**Education level**				
High	3.34	1.07–10.4 *	2.53	0.88–7.72
Low	1		1	
**Working hour**				
Less than 8 working hour	1		1	
More than 8 working hour	6.99	1.97–24.8 *	2.53	0.83–7.72
**Emotional support**				
High	6.77	0.96–47.6	2.11	0.55–19.5
Low	1		1	
**Informational support**				
High	0.43	0.96–2.82	0.39	0.06–2.52
Low	1		1	
**Instrumental support**				
High	1.55	0.19–12.5	1.71	0.22–13.0
Low	1		1	
**Appraisal support**				
High	1.44	0.19–10.7	2.92	0.39–21.9
Low	1		1	

RRR: Risk Relative Ratio with the reference of exclusive breastfeeding. * Significantly statistic *p* value less than 0.05.

## Data Availability

All of the primary data and materials involved in this paper are from the primary data source. If readers need more information about data and materials, please contact author for data requests.

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
