# Peer review of "Analysis of Supporting Factors Associated with Exclusive Breastfeeding Practice in the Urban Setting during the COVID-19 Pandemic"

_children, 2022, doi:10.3390/children9071074_

Round 1
Reviewer 1 Report
INTRODUCTION
with a breastfeeding rate of 27.5%, and only 42% of children under six months are exclusively breastfed, respectively.
Comment: contradiction between (initial?) breastfeeding rate of 27.5% and exclusive breastfeeding rate of 42% at 6m.
Please translate reference 17 to English.
Table 1. Mother age group, please convert month to years.
Table 2, multivariable regression does not support these CONCLUSIONS: Mother's factors such as age, education, level of education support the success of breastfeeding in the first six months of birth.
Author Response
Responses to the Reviewers' comments # 1
Response:
We appreciate very much the reviewers for the effort and time put into the review of the manuscript. Each comment has been carefully considered point by point and responded to your comments. Responses to the reviewers and changes in the revised manuscript are as follows.
|
S. N |
Comments from reviewer 1 |
Changed to |
|
1 |
with a breastfeeding rate of 27.5%, and only 42% of children under six months are exclusively breastfed, respectively. Comment: contradiction between (initial?) breastfeeding rate of 27.5% and exclusive breastfeeding rate of 42% at 6m.
|
Thank you for the review. Along with the manuscript it has been changed appropriately. |
|
2 |
Please translate reference 17 to English. |
Thank you for the comment. We have translated regarding that reference. |
|
3 |
Table 1. Mother age group, please convert month to years. |
We have edited Table 1. The month word has been changed to years |
|
4 |
Table 2, multivariable regression does not support these CONCLUSIONS: Mother's factors such as age, education, level of education support the success of breastfeeding in the first six months of birth |
We agree with your comment. We have reanalyzed our data and we have change the Table 2. So, our conclusion has been made following the current result.
|

Reviewer 2 Report
In my opinion the article needs to be revised. I would greatly recommend the authors to rewrite the Results section. Otherwise is difficult to following the study and understand the conclusions. In this section, the authors begin by talking about some of the results (percentages) presented in table 1. In that table they also presented the p values. However, they don´t give any information about these p values. What was the statistical analysis that was performed? What are the conclusions? A more complete information is required. In table 1, the age of mothers is in months. Please correct
The qualitative results are mixed with the quantitative results, table 2 appears in the in the middle of the interviews. In my opinion the qualitative and quantitative data must be presented separately, one subsection for each. I would greatly recommend the authors to put as much information as possible regarding to statistical analysis in table 2. They should inform that the interpretation of the multinomial logistic regression in given in terms of relative risk ratios. I would greatly recommend the authors to provide more information about the results provided in table 2.
Regarding to statistical analysis the authors should give information about the assumptions of the methods used. They also should give information about model fitting.
The conclusions section needs to be improved.
The authors should give more information about The Family Support Questionnaire.
Author Response
Responses to the Reviewers' comments # 2
Response:
We appreciate very much the reviewers for the effort and time put into the review of the manuscript. Each comment has been carefully considered point by point and responded to your comments. Responses to the reviewers and changes in the revised manuscript are as follows.
|
S. N |
Comments from reviewer 2 |
Changed to |
|
1 |
a. In my opinion the article needs to be revised. I would greatly recommend the authors to rewrite the Results section. Otherwise is difficult to following the study and understand the conclusions. b. In this section, the authors begin by talking about some of the results (percentages) presented in table 1. In that table they also presented the p values. However, they don´t give any information about these p values. What was the statistical analysis that was performed? What are the conclusions? A more complete information is required. c. In table 1, the age of mothers is in months. Please correct |
a. Thank you for the review. Along with the manuscript it has been changed appropriately. b. We agree with reviewer point of view for this issues. We have added basic principle of statistical analysis briefly in the methods section line 100-105 c. We have edited Table 1. The age of mothers has been changed from month to years old (y.o). |
|
2 |
a. The qualitative results are mixed with the quantitative results, table 2 appears in the in the middle of the interviews. In my opinion the qualitative and quantitative data must be presented separately, one subsection for each. b. I would greatly recommend the authors to put as much information as possible regarding to statistical analysis in table 2. c. They should inform that the interpretation of the multinomial logistic regression in given in terms of relative risk ratios. I would greatly recommend the authors to provide more information about the results provided in table 2.
|
a. Thank you for the comment. We have agreed with this statement. The quantitative and qualitative parts have been presented separately in the result sections as you suggested. b. We have updated the procedure as well as other information related with statistical analysis. We also have reanalyzed the Table 2 accordingly. c. We surely agreed with this statement. We have added your suggestion on the result and discussion part. |
|
3 |
Regarding to statistical analysis the authors should give information about the assumptions of the methods used. They also should give information about model fitting. |
Thank you for the comment. The reason why we decide to use the multinomial regression in statistical analysis due to we have three level of dependent variable (breastfeeding, infant formula milk, and a combination of breastfeeding and infant formula milk). Regarding with the information about model fitting, we only input the independent variable with the pvalue less than 0.20 resulted from chisquare test (Table 1) into the multinomial regression (Table 2) |
|
4 |
The conclusions section needs to be improved. |
Thank you for the comment. We agree with your comment. |
|
5 |
The authors should give more information about The Family Support Questionnaire.
|
Reference related with the Family Support Questionare is already added. Originally it is based on doctoral dissertation and already translated into Indonesian.
|

Round 2
Reviewer 1 Report
The manuscript has been changed appropriately.
Author Response
We appreciate very much the reviewers for the effort and time put into the review of the manuscript. Thank you for your suggestion that making the current version is appropriate for publication.

Reviewer 2 Report
I appreciate the great work, but I would like to suggest the following points to strengthen your paper:
Comments:
Lines 97-98
The chi-square test for independence compares two variables categorical variables in a con to see if a contingency table to see if they are related. That’s your case, the variables in table 1 are categorical so I suggest the following
Lines 97-98: Firstly, the chi-square test for independence was used to test the association between sociodemographic variables and breastfeeding practice patterns, then the multinomial logistic regression analysis…..
- Table 1 : in the table header the “n” of MIXED is out of place.
Dear authors by looking at the results (p-value) at table 1, we conclude that there is association between sociodemographic variables and breastfeeding practice patterns. So you must report that in your results.
I recommend that the manuscript be accepted after the comments above are addressed
Author Response
We appreciate very much the reviewers for the effort and time put into the review of the manuscript. The changes in the second revised manuscript was highlighted by blue color. Each comment has been carefully considered point by point and responded to your comments as follows.
|
S. N |
Comments from reviewer 2 |
Changed to |
|
1 |
Lines 97-98 |
Thank you for the suggestion in making our paper has more strengths. We agree with this suggestion. We then, along with the manuscript has been changed appropriately in the lines 97-98 as suggested. |
|
2 |
a. Table 1: in the table header the "n" of MIXED is out of place. b. The results (p-value) at table 1, we conclude that there is association between sociodemographic variables and breastfeeding practice patterns. So, you must report that in your results. |
a. Thank you for the comment. We have agreed with this statement. we have replaced the header “n” of MIXED appropriately. b. We have added “there is association between sociodemographic variables and breastfeeding practice patterns” in the paragraph of result section as suggested. |
